# The Clinical Significance of *CRNDE* Gene Methylation, Polymorphisms, and CRNDEP Micropeptide Expression in Ovarian Tumors

**DOI:** 10.3390/ijms25147531

**Published:** 2024-07-09

**Authors:** Laura Aleksandra Szafron, Roksana Iwanicka-Nowicka, Agnieszka Podgorska, Arkadiusz M. Bonna, Piotr Sobiczewski, Jolanta Kupryjanczyk, Lukasz Michal Szafron

**Affiliations:** 1Maria Sklodowska-Curie National Research Institute of Oncology, 02-781 Warsaw, Poland; 2Laboratory of Systems Biology, Faculty of Biology, University of Warsaw, 02-106 Warsaw, Poland; 3Laboratory for Microarray Analysis, Institute of Biochemistry and Biophysics, Polish Academy of Sciences, 02-106 Warsaw, Poland; 4Cancer Molecular and Genetic Diagnostics Department, Maria Sklodowska-Curie National Research Institute of Oncology, 02-781 Warsaw, Poland; 5Triple Helical Peptides Ltd., Cambridge CB22 5DU, UK; abonna@peptidebiochem.eu; 6Department of Gynecological Oncology, Maria Sklodowska-Curie National Research Institute of Oncology, 02-781 Warsaw, Poland; 7Department of Cancer Pathomorphology, Maria Sklodowska-Curie National Research Institute of Oncology, 02-781 Warsaw, Poland; jolanta.kupryjanczyk@nio.gov.pl

**Keywords:** ovarian cancer, lncRNA, DNA sequence variant, DNA methylation, IHC, micropeptide, *CRNDE*, CRNDEP, TP53

## Abstract

*CRNDE* is an oncogene expressed as a long non-coding RNA. However, our team previously reported that the *CRNDE* gene also encodes a micropeptide, CRNDEP. The amino acid sequence of CRNDEP has recently been revealed by other researchers, too. This study aimed to investigate genetic alterations within the CRNDEP-coding region of the *CRNDE* gene, methylation profiling of this gene, and CRNDEP expression analysis. All investigations were performed on clinical material from patients with ovarian tumors of diverse aggressiveness. We found that CRNDEP levels were significantly elevated in highly aggressive tumors compared to benign neoplasms. Consistently, a high level of this micropeptide was a negative, independent, prognostic, and predictive factor in high-grade ovarian cancer (hgOvCa) patients. The cancer-promoting role of CRNDE(P), shown in our recent study, was also supported by genetic and epigenetic results obtained herein, revealing no CRNDEP-disrupting mutations in any clinical sample. Moreover, in borderline ovarian tumors (BOTS), but not in ovarian cancers, the presence of a single nucleotide polymorphism in *CRNDE*, rs115515594, significantly increased the risk of recurrence. Consistently, in BOTS only, the same genetic variant was highly overrepresented compared to healthy individuals. We also discovered that hypomethylation of *CRNDE* is associated with increased aggressiveness of ovarian tumors. Accordingly, hypomethylation of this gene’s promoter/first exon correlated with hgOvCa resistance to chemotherapy, but only in specimens with accumulation of the TP53 tumor suppressor protein. Taken together, these results contribute to a better understanding of the role of CRNDE(P) in tumorigenesis and potentially may lead to improvements in screening, diagnosis, and treatment of ovarian neoplasms.

## 1. Introduction

Ovarian carcinoma (OvCa) is still the leading cause of death in women diagnosed with female genital tract cancers [1] despite the development of new therapies and a better understanding of cancer biology. Recently, *CRNDE* has been one of the most extensively studied genes in many types of tumors [2,3]. It is considered to be an oncogene, which codes for a long non-coding RNA (lncRNA), though our team also discovered the 84-amino-acid micropeptide, CRNDEP (GenBank acc. no. ACJ76642.1), encoded by one of the *CRNDE* transcripts (GenBank acc.no. FJ466686.1) [4]. In the same paper, we demonstrated predominantly nuclear localization of CRNDEP, while its expression was increased in highly proliferating tissues, both normal and malignant, thus implying the involvement of this micropeptide in the positive regulation of the cell cycle. Consistently, in our newest study [5], we proved the oncogenic potential of *CRNDE* and CRNDEP, demonstrating their multi-faceted impact on carcinogenesis at the stages of DNA transcription and replication, RNA metabolism, and also cell cycle progression and proliferation. This study involved the whole transcriptome sequencing of SK-OV-3 ovarian cancer cells after the sh-RNA-induced silencing of *CRNDE*, and various in vitro experiments on SK-OV-3, A2780, and HeLa cells with either elevated or diminished *CRNDE* expression. In the same paper, we revealed the presence of CRNDEP in both the centrosomes of dividing cells and in nucleoli, where it can interact with p54, an RNA helicase. Furthermore, we proved that CRNDE(P) expression changes affected the activity of the microtubular cytoskeleton and the formation of focal adhesion plaques, while high CRNDE(P) expression was found to increase the resistance of OvCa cells to microtubule-targeted cytostatics [5]. In another study, we established that increased expression of lncRNA-coding and CRNDEP-coding transcripts of *CRNDE* alike was a negative prognostic factor in OvCa patients [6]. Although the role of *CRNDE* lncRNA is quite well understood in ovarian cancer, especially in cell lines [7,8], the status of *CRNDE* gene methylation, as well as the role of the CRNDEP micropeptide are still poorly recognized. Therefore, this study aimed to broaden the knowledge on the role of the *CRNDE* gene and its protein product, CRNDEP, in ovarian tumors of different aggressiveness, i.e., the most aggressive high-grade ovarian carcinomas (hgOvCa), borderline ovarian tumors (BOTS) without (BOT) and with the *BRAF* V600E mutation (BOT V600E), and low-grade ovarian carcinomas (lgOvCa). To achieve this goal, we looked for genetic alterations within the CRNDEP-coding region of *CRNDE*, profiled DNA methylation in this entire gene, and analyzed the differential expression of the CRNDEP micropeptide. These three experiments were performed on a retrospective cohort of patients with ovarian tumors in the context of detailed clinicopathological data.

## 2. Results

### 2.1. Analysis of Genetic and Epigenetic Alterations within the CRNDE Gene in Ovarian Tumors

A complete coding sequence of the *CRNDE* gene (NM_001308963) was searched for genetic alterations, potentially affecting the structure and/or function of the CRNDEP micropeptide, in the entire set of 225 ovarian tumors investigated herein. Interestingly, we found no insertions or deletions within this region. As to single nucleotide polymorphisms (SNPs) within the CRNDEP-coding region, only one genetic alteration was found, i.e., GRCh38:chr16:g.54919202G>A, NP_001295892.1:p.Pro47Leu, rs115515594 (Appendix A). This missense variant was identified in two BOT, three BOT V600E, no lgOvCa, and two hgOvCa. Its presence does not seem to make an impact on the CRNDEP micropeptide level in the cells, which aligns with the bioinformatic predictions shown in our previous research, suggesting that Pro47 lies within an unstructured region of CRNDEP [4]. Nevertheless, the aforementioned genetic alteration in *CRNDE* occurred more frequently in BOT V600E tumors than in hgOvCa, and this difference was statistically significant (two-sided Fisher’s exact test: OR 10.0, 95% CI 1.08–126.9, *p* = 0.021). Moreover, the same SNP significantly increased the risk of recurrence in the entire BOTS series (HR 14.63, 95% CI 1.948–109.872, *p* = 0.0091). Notably, the rs115515594 variant exhibited its adverse clinical meaning independently of the clinical stage of the tumor and its histological type, both of which were also identified as prognostic factors in this multivariable analysis (Figure 1). Moreover, the analyzed *CRNDE* polymorphism affected BOTS prognosis irrespective of whether tumors harbored the *BRAF* V600E mutation or not, while this *BRAF* alteration made no impact on the clinical outcome in BOTS.

To further investigate the role of the aforementioned SNP in BOTS pathogenesis, we checked the distance between the *CRNDE* locus on chromosome 16 and the loci of other genes located on chromosome 16, known to be associated with ovarian cancer development (*TSC2*, *PALB2*, and *FANCA*). All these genes are included in one of two NGS panels, used in our upcoming study (Szafron LA. et al., the paper is currently being prepared for publication), offering the enrichment of genes involved in hereditary ovarian carcinoma development (Ion AmpliSeq™ Comprehensive Ovarian Cancer Research Panel, Thermo Fisher Scientific, abbreviated as Thermo, Waltham, MA, USA) or frequently mutated in sporadic human cancers (KAPA HyperPETE Hot Spot Panel, Roche, Basel, Switzerland). The distance in centimorgans (cM), measured with the cM Estimator (https://dnapainter.com/tools/cme) between *CRNDE* and the *FANCA*, *PALB2* and *TSC2* genes equaled 63.3 cM, 24.5 cM, and 66.2 cM, respectively. Thus, the risk of linkage had to be assessed for the *PALB2* gene only (the distance between *CRNDE* and *PALB2* was smaller than 50 cM). To check whether the presence of genetic variants in *PALB2* affects the importance of the rs115515594 SNP in *CRNDE*, a multivariable regression analysis was carried out. This inference revealed no impact of genetic variants in *PALB2* on the clinical outcome in BOTS, concomitantly proving that rs115515594 can be considered an independent marker of poor prognosis in borderline ovarian tumors (Appendix A).

Additionally, we looked for other genetic variants being in linkage disequilibrium (LD) with rs115515594 in the European population of healthy individuals (Appendix A). This approach let us identify 40 other SNPs. Interestingly, all of them were located in non-coding regions of the genome, outside the *CRNDE* locus, and none was highly correlated with the polymorphism identified here as a potential prognostic biomarker in BOTS (the maximum value of the coefficient of determination (r^2^) equaled 0.319). Furthermore, no rs115515594-linked SNPs were listed in the ClinVar database. Of note, there was only one polymorphism in LD with rs115515594, rs117633800 (located within the *CRNDE* promoter, approximately 1400 bp upstream from the transcription start site), that is likely to play regulatory roles in the cells, as assessed based on the FORGEdb and RegulomeDB scores. The rs117633800 variant was present in all our tumors harboring rs115515594. Thus, the occurrence of these two SNPs seems more highly correlated in ovarian tumors than in the European population of healthy individuals. To determine if both SNPs occur more often in our tumor set compared to normal DNA samples of European origin, we evaluated their frequencies with the Fisher exact test. This inference revealed statistically significant overrepresentation, stronger for rs115515594 and weaker for rs117633800 in BOTS (odds ratio (OR) 17.52, *p*-value = 0.00015; OR 2.93, *p*-value = 0.0294, respectively). By contrast, in OvCa, the prevalence of these two SNPs was similar to that observed in healthy Europeans. Based on these results, one may assume that the presence of the rs115515594 SNP within the CRNDEP-coding sequence is crucial for BOTS, concomitantly playing no role in OvCa development and progression. Furthermore, the rs115515594 variant turned out to affect prognosis independently of polymorphisms in other genes determining clinical outcome in BOTS (these genes will be described in our next paper by Szafron LA. et al., currently being prepared for publication). All the results suggest that the clinical impact of the rs115515594 SNP may be attributed to this genetic alteration with relatively high confidence. Yet, it needs to be emphasized that this polymorphism remains in linkage disequilibrium with rs117633800 and 39 other, potentially functional SNPs (undetectable in our current experimental setup), the presence of which could also contribute to BOTS tumorigenesis, either by augmenting or reducing the effects shown in this research.

The differential methylation analysis of the *CRNDE* gene was carried out in serous ovarian tumors not only for the individual CpG sites but also in a broader context of entire gene regions (according to annotations provided by the FANTOM database, available at https://fantom.gsc.riken.jp/ (accessed on 20 November 2021)) and so-called differentially methylated regions (DMRs), being gene-independent areas of the genome, defined by the density of CpGs with statistically significant methylation differences between the analyzed groups of samples. From among six distinct regions of the *CRNDE* gene subjected to the CpG methylation analysis in the present study (distal promoter (1to5kb), proximal promoter, first exon, introns, intron/exon boundaries, lncRNA), the distal promoter was the only region showing no differences in methylation patterns between BOT, BOT V600E, lgOvCa, and hgOvCa. In all the other regions, we found CpG hypomethylation to be positively correlated with higher tumor aggressiveness (Figure 2). Of note, this regularity was most evident in the first exon (Kruskal–Wallis (KW) test *p* = 1.011 × 10^−9^) and the lncRNA-coding region (KW test *p* = 1.512 × 10^−8^) of the *CRNDE* gene (Figure 2A,B).

Apart from the CpG-oriented analysis, we also identified one DMR overlapping the *CRNDE* gene (GRCh37:chr16:54962001-54963401), encompassing its promoter, first exon, and first intron (Figure 3A). Our statistical inference revealed that the differences in the methylation pattern within this region were significant for BOT vs. hgOvCa comparison only, proving hypomethylation of *CRNDE* in more aggressive tumors. This DMR on chromosome 16 comprised 1401 bp and contained nine CpG sites (Figure 3B), six of which were located on the minus (*CRNDE*-coding) DNA strand (g.54962001, g.54962423, g.54962487, g.54962832, g.54963240, g.54963401). On the opposite (plus) DNA strand, only three CpGs were forming this DMR (g.54962385, g.54962996, g.54963224). To evaluate whether the analysis of methylation patterns in the identified DMR can be of potential clinical use, multivariable Cox and logistic regression analyses were carried out. According to their results, the *CRNDE* methylation changes did not affect BOTS patients’ relapse-free survival (RFS) and aggressiveness of their tumors (the presence of microinvasions or implants). In hgOvCa, methylation alterations within the studied DMR did not make an impact on cancer prognosis as well (patient overall survival (OS) and disease-free survival (DFS)). Nevertheless, this DMR may probably serve as a marker predictive of a good response to chemotherapy, since higher methylation level in this region increased the sensitivity of hgOvCa to chemotherapy, independently of the residual disease (being an adverse predictive factor) (Figure 4). Notably, this regularity was observed only in hgOvCa with accumulation of the TP53 protein, irrespective of whether the DMR with CpG sites located on both DNA strands (OR 2.935, 95% CI 1.062–8.11, *p* = 0.038) or on the minus strand only (OR 2.862, 95% CI 1.065–7.693, *p* = 0.037) were taken into account.

Finally, in order to verify the reliability of the observed methylation alterations, we selected a single CpG site within the first intron of the *CRNDE* gene with a diverse methylation pattern in our microarray analysis, i.e., cg13471560, GRCh37:chr16:54960485 (Figure 3A,C). Methylation of this CpG site was then assessed for three samples with methylation-specific PCR followed by Sanger sequencing. The sequencing results (Figure 3D) fully aligned with the microarray data, thereby validating the computational algorithms used herein for the methylation analysis.

### 2.2. Investigating the Expression of CRNDEP in Ovarian Tumors

Before the CRNDEP expression changes were assessed in our cohort of snap-frozen tumor samples by dot blot, the quality of the anti-CRNDEP antibody was evaluated with immunohistochemical (IHC) staining for 10 tumors characterized by different aggressiveness. Exemplary results depicting the moderate and strong nuclear expression of CRNDEP are shown in Figure 5A,B. To, once again [4], prove the specificity of this staining, the complete lack of IHC reaction in the tumor exhibiting strong CRNDEP expression, triggered by a blocking peptide, is demonstrated in Figure 5C. As the blocking peptide, a synthetic CRNDEP epitope (employed previously to develop the anti-CRNDEP antibody) was used.

In the dot blot analysis, we investigated the differences in CRNDEP levels between the analyzed groups of neoplasms and established that the expression of this micropeptide in serous ovarian tumors tends to increase gradually with their elevating aggressiveness (see mean CRNDEP expression values in Figure 5D). However, this difference is statistically significant only in the comparison between BOT and hgOvCa (Wilcoxon rank sum test *p*-value = 0.0213). The lack of relevant differences in CRNDEP expression in the remaining comparisons might be due to the relatively low number of specimens in the non-hgOvCa groups.

Next, multivariable Cox and logistic regression analyses were performed to assess the clinical importance of CRNDEP expression in BOTS and hgOvCa. Although we found no relationship between CRNDEP levels and the clinical outcome in BOTS, high expression of this micropeptide emerged as a negative prognostic and predictive factor in hgOvCa (Table 1 and Figure 6). Concerning cancer prognosis, high levels of CRNDEP increased the risk of death. This adverse impact was the strongest in the subgroup of patients treated with the taxane/platinum (TP) regimen (HR 6.388, 95% CI 2.0–21, *p* = 0.0020). CRNDEP emerged as a prognostic marker independent of the high clinical stage (FIGO) and residual tumor size (RT), both of which were identified as predictors of poor prognosis. Consistently, the negative prognostic meaning of high CRNDEP expression was also discovered in the entire set of hgOvCa specimens, a subset with the TP53 protein accumulation, and the TP-treated subgroup with TP53 accumulation. As to the prediction of treatment response, high CRNDEP expression diminished the chances of both hgOvCa tumors’ complete remission (CR), and their sensitivity to chemotherapy (PS), thus supporting the outcome of our recently published in vitro experiments [5]. Similarly to the analysis of prognosis, this adverse clinical effect was observed in the entire cohort of hgOvCa patients (CR: OR 0.564, 95% CI 0.4–0.9, *p* = 0.0067; PS: OR 0.603, 95% CI 0.4–0.9, *p* = 0.0162), and in those subjected to the TP treatment (CR: 0.531, 95% CI 0.3–0.8, *p* = 0.0053; PS: 0.591, 95% CI 0.4–0.9, *p* = 0.0160), but not in the platinum/cyclophosphamide (PC)-treated individuals. This predictive value of CRNDEP expression turned out to be independent of the big RT size, being a negative predictive factor, too. Interestingly, a negative predictive meaning of CRNDEP expression was also revealed for CR in the subgroups of hgOvCa with the TP53 protein accumulation.

### 2.3. Evaluating the Correlation between the CRNDE Gene Methylation, Expression, and Levels of CRNDEP in OvCa

To investigate the relationship between the *CRNDE* gene methylation, the expression of the CRNDEP-coding transcript, and CRNDEP levels in corresponding 92 OvCa samples, a Spearman’s correlation matrix was created. As presented in Appendix A, there was no statistically significant correlation between methylation alterations and either the expression of the CRNDEP-coding transcript or CRNDEP levels. By contrast, the expression of the *CRNDE* gene and the levels of the CRNDEP micropeptide were found to be mutually positively correlated (*r* = 0.21, *p* < 0.05).

## 3. Discussion

Herein, to investigate the clinical usability of the *CRNDE* gene and the CRNDEP micropeptide, we assessed DNA methylation patterns in the whole *CRNDE* locus, looked for genetic alterations in the CRNDEP-coding region of this gene, and investigated how CRNDEP level differed between ovarian tumors of diverse aggressiveness. To evaluate the prognostic and predictive value of these potential biomarkers, all analyses were performed in the context of detailed clinicopathological data. Our multivariable regression analyses confirmed the oncogenic role of the *CRNDE* gene and its micropeptide product in hgOvCa, whereas in lgOvCa, we could not perform such statistical inference due to the low number of specimens in this set of tumors. By contrast, in BOTS, no clinical value of either altered *CRNDE* methylation or changed CRNDEP expression was found in the present study. However, we established that the only genetic variant altering the CRNDEP sequence, identified in our research, rs115515594, occurred statistically more frequently in BOT V600E tumors than in hgOvCa. Consistently, the presence of this SNP in *CRNDE* significantly increased the risk of recurrence in BOTS, in which this genetic variant was highly overrepresented compared to healthy individuals. Notably, the clinical value of the rs115515594 SNP was not found in hgOvCa.

To begin with, we decided to carry out all our investigations presented in this article in two subsets of BOTS, one with the *BRAF* V600E mutation and another lacking this genetic alteration. The main reason for this division is that our preliminary analyses revealed that BOT V600E tumors are diagnosed in younger patients than BOTS lacking this genetic alteration (Appendix A). Furthermore, our yet unpublished results of the whole methylome analysis in BOTS and OvCa uncovered that, between BOT V600E and BOT, there are over 20,000 differentially methylated CpG sites (which made these two subgroups more diverse than when BOT V600E were compared to either lgOvCa or hgOvCa). Of note, according to the literature, mutations in the *BRAF* oncogene are typical of BOTS but not of OvCa [9,10]. Of all alterations in this gene, the V600E variant is most frequent (accounting for over 90% of *BRAF* mutations), and genetic changes in this codon (classified as class I mutations) result in the continuous activation of the *BRAF* kinase, which, acting as a monomer, exhibits stronger ERK phosphorylation activity than class II and III *BRAF* mutations [11]. This leads to the constant transduction of pro-proliferative signals regardless of external stimuli, eliciting a cancer-driving effect [11]. Given the distinctive characteristic of mutations in the *BRAF* V600 codon, their high clinical impact, and the lack of other *BRAF* V600 variants in our BOTS, we decided to divide this set of tumors into two categories depending on the presence or absence of the *BRAF* V600E genetic lesion in their genomes.

As to sequence variants in the *CRNDE* gene, we observed the lack of CRNDEP-disrupting mutations (there were no non-SNP variants and no SNPs affecting either the start/stop codon or splice sites) in our entire set of tumors. The only missense variant found, rs115515594, leads to the substitution (Pro47Leu) of proline, an exceptionally rigid amino acid strongly affecting the secondary structure of a protein [12], with non-polar leucine. Although the Ensembl database classifies this change as exerting a moderate impact on the protein’s structure/function, an analogical substitution was previously shown to impair the function of guanylate cyclase-activating protein-1, leading to its lower stability and cell concentration, which resulted in cone-rod dystrophy [13]. Another study demonstrated that an opposite substitution (leucine with proline) in caspase-9 may lead to neuroblastoma development [14]. By contrast, in the present study, we did not observe altered CRNDEP levels in tumors with the CRNDEP Pro47Leu substitution. This is likely caused by the fact that the analyzed SNP is located in the region coding for an unstructured fragment of CRNDEP [4], and, therefore, its presence does not affect the stability of this micropeptide. Nevertheless, we demonstrated here that the rs115515594 SNP is a factor of poor prognosis, though such regularity was found only in BOTS, and not in hgOvCa. Consistently, the same genetic variant occurred more frequently in BOT V600E tumors than in highly aggressive OvCa. This outcome suggests the mechanisms driving benign and malignant ovarian neoplasms to be different, which was also corroborated by other researchers [15,16]. Still, the molecular background underlying the adverse prognostic impact of the rs115515594 SNP in BOTS remains unknown and needs to be addressed in future studies.

The oncogenic role of *CRNDE* also emerged in our DNA methylation analysis, as *CRNDE* hypomethylation in many regions of this gene correlated with increased tumor aggressiveness. This regularity was the strongest when the most advanced carcinomas (hgOvCa) were compared to BOTS and lgOvCa. Remarkably, the proximal promoter region of *CRNDE* was characterized by significantly lower methylation in lgOvCa than in BOTS, whereas no methylation changes within the *CRNDE* gene were found in any inter-BOTS comparison. Such a result is consistent with the current knowledge, as global, genome-wide hypomethylation and diminished methylation of oncogenes in advanced carcinomas is a well-known phenomenon [17]. Interestingly, the proximal promoter and the 1st exon of *CRNDE* were the regions strongly hypomethylated in our hgOvCa group. In line with this finding, hypomethylation of a DMR encompassing these two regions of *CRNDE* increased hgOvCa resistance to chemotherapy in our study. Changes in promoter methylation are generally acknowledged as a gene expression-changing factor, whereas promoter down-methylation in various oncogenes was linked to lower sensitivity of neoplastic cells to chemotherapy. For example, in OvCa, *TMEM88* promoter hypomethylation was associated with platinum resistance [18]. Similarly, elevated myelin and lymphocyte (MAL) oncogenic protein expression was linked to promoter hypomethylation of the corresponding gene, and platinum resistance in epithelial OvCa [19]. Increased methylation of the first exon of a gene is also tightly connected to its transcriptional silencing, conceivably even stronger than that caused by hypermethylation of the upstream-lying promoter [20]. Nonetheless, no scientific reports on a relationship between the first exon methylation alterations and OvCa chemotherapy resistance are currently available. It needs to be emphasized that in our regression analyses, we confirmed the adverse clinical meaning of the *CRNDE* promoter/1st exon hypomethylation in tumors with the TP53 protein accumulation only. The TP53 accumulation, which results mainly from the presence of missense mutations in the *TP53* gene, is one of the most frequent genetic aberrations occurring in OvCa. Such mutations exert a dominant-negative effect leading to the impairment of the TP53 tetramer formation and the loss of a normal tumor-suppressive function of this protein [21,22]. Notably, our previous studies showed that the presence of TP53 accumulation in the cell, resulting from the inactivation of the TP53 tumor suppressor, creates an environment in which the clinical meaning of other molecular markers, such as BCL2, BAX, ERBB2 [23,24], or p19INK4d [25], can be unraveled. Moreover, we previously proved that accumulation of the TP53 protein correlated with decreased expression of *CRNDE* transcripts in hgOvCa [6]. The results presented herein seem to align with that discovery, since increased methylation of *CRNDE* (that, according to another study on chronic lymphocytic leukemia [26], correlates with decreased *CRNDE* expression) was associated with higher sensitivity of cancer cells to chemical treatment. Of note, this favorable clinical outcome was observed only in hgOvCa with TP53 accumulation. This may appear illogical at first glance, implying that in cells with impaired TP53, the activity of the *CRNDE* oncogene is silenced. However, our recent study shed new light on this intriguing phenomenon, revealing that a stable knockdown of *CRNDE* in ovarian cancer cells harboring a homozygous null TP53 mutation (SK-OV-3) resulted in massive up-regulation of hundreds of genes, and their number was significantly higher than those which were down-regulated [5]. Such an effect is conceivably related to the aberrant action of the polycomb repressive complex 2 (PRC2), which maintains overall transcriptional repression in the genome, and interacts with *CRNDE* as previously shown [27]. So, down-regulation of *CRNDE* in OvCa tumors harboring TP53 accumulation may be a mechanism by which a cancer cell tries to activate many oncogenes at once, thus boosting its aggressiveness in the absence of functional TP53.

Next, in the present paper, we not only corroborated the oncogenic role of the *CRNDE* gene but also obtained results on clinical material supporting the outcome of our recent in vitro study [5], that portrayed CRNDEP as a molecule eliciting a similar, cancer-driving effect. CRNDEP belongs to small open reading frame (smORF)-encoded polypeptides (SEPs). The median length of SEPs identified so far is only 50 amino acids, which suggests that they probably do not exhibit enzymatic activity, but only modulate the activity of proteins to which they bind, or play the role of signaling molecules [28,29], such as myoregulin and sarcolipin [30]. Other studies have shown the impact of micropeptides on DNA repair, inflammation, metabolism, and carcinogenesis [31]. In recent years, a few research teams independently confirmed that the *CRNDE* transcript (recognized by us as CRNDEP-coding) can bind ribosomes along its entire length. Furthermore, the specificity of this binding indicated translation [32,33,34]. Despite many technical difficulties, the CRNDEP existence was recently revealed in mass spectrometry analyses by our team [5] and by others [31]. The trickiness of CRNDEP identification was due to its instability upon exposure to various physicochemical factors (high temperature, sodium dodecyl sulfate, dithiothreitol, Laemmli buffer) [5]. This is not surprising, as micropeptides are usually rather labile and are present in small amounts in the cell. Additionally, they may also exhibit tissue- and time-specific patterns of expression, which also impedes their identification and analysis [29]. The aforementioned lability made it impossible to detect CRNDEP in its endogenous form by Western blot or ELISA. Therefore, we had to rely on either dot-blot or IHC to assess the amount of this micropeptide in tumor cells. Both these techniques were used herein to optimize the analytical workflow. First, IHC staining with a blocking peptide was utilized to, once again [4], corroborate the specificity of the anti-CRNDEP antibody. Afterward, we applied the dot blot technique to assess CRNDEP expression in all our snap-frozen tumor specimens. If the stromal cell contamination (scc) value is precisely evaluated for every sample (like herein), dot blot outperforms IHC in two main ways, not only in terms of the speed of the analysis but also, and foremost, by offering a truly quantitative and unbiased outcome, which fully reflects the mean expression of the analyzed protein in every sample. By contrast, in IHC, focal staining reaction and problems with staining of old formalin-fixed paraffin-embedded (FFPE) material (an old retrospective cohort of patients was used in our study) may occur. Consistently, Grillo et al. [35] indicated that membrane and nuclear (CRNDEP belongs to this group) antigens presented reduced staining intensity in older paraffin blocks, while cytoplasmic antigens showed no reduction in immunostaining intensity over time. Our dot blot analysis revealed that in highly aggressive ovarian tumors (hgOvCa), CRNDEP level was significantly increased compared to BOTS without the *BRAF* V600E mutation (BOT). This result perfectly aligns with the outcome of our earlier study [4], where we concluded that CRNDEP expression is elevated in highly proliferating cells, normal and malignant alike. It seems intriguing that herein mean CRNDEP expression tended to increase gradually with rising aggressiveness of the tumor cells, though the remaining inter-group comparisons turned out insignificant, likely due to insufficient sample sizes. Accordingly, a high level of CRNDEP emerged as a negative prognostic factor, increasing the risk of death in hgOvCa patients, as well as the factor of poor response to the TP chemotherapy regimen. Remarkably, the impact of chemotherapy on patient survival appears particularly interesting, given that the adverse clinical meaning of CRNDEP expression was the strongest in the TP-treated patients. Taxanes are cytostatic drugs that stabilize the microtubule cytoskeleton [36]. This structure plays a central role in both cell division and intracellular trafficking of DNA repair proteins [37]. Notably, in our recently published study [5], we demonstrated in vitro, in two different ovarian cancer cell lines, that elevated CRNDE(P) expression confers resistance of the cells to several microtubule-targeting agents, including paclitaxel. This phenomenon would explain both the greatly unfavorable impact of CRNDEP overexpression in tumors of TP-treated individuals on their survival, and also the diminished sensitivity of hgOvCa with high CRNDEP levels to this type of chemotherapy.

Lastly, by showing the correlation between the expression of the CRNDEP-coding transcript and the micropeptide itself, we managed to show the cause–effect relationship between the levels of these two biomolecules. This result appeared predictable, given that high levels of CRNDEP and *CRNDE* transcripts alike were demonstrated to be predictors of poor clinical outcomes in OvCa patients herein and in our previous study [6]. Nevertheless, the expression correlation, though statistically significant, was characterized by a relatively low correlation coefficient (r) value. This suggests that, though reciprocally, positively linked, the levels of CRNDEP and its transcript do not change linearly in the cells (hence, the low r value). According to the literature, the levels of gene transcripts and the corresponding proteins are not always concordant, due to various regulatory mechanisms, affecting the stability and degradation rates of both RNA and proteins [38]. Yet, it was established in ovarian cancer xenograft models that the transcript–protein expression correlation was far better for differentially expressed mRNAs than those lacking differential expression [39]. Notably, this outcome is supported by the results presented herein. On the other hand, the absence of correlation between the *CRNDE* gene promoter/first exon methylation and its mRNA expression, shown in Appendix A, seems discordant with the generally acknowledged expression-silencing impact of DNA hypermethylation reported by others [40]. This inconsistency is likely caused by the fact that *CRNDE*, apart from a micropeptide, also encodes lncRNA, which was demonstrated to interact with the polycomb repressive complex 2 (PRC2) [27]. This epigenetically active complex maintains overall transcriptional repression in the genome [41]. The *CRNDE* gene knockdown presented in our previous study on ovarian cancer cell lines seems to confirm its relationship with PRC2, since, after *CRNDE* silencing, the number of genes with elevated expression in SK-OV-3 cells was significantly higher than those with reduced expression [5]. Given the involvement of *CRNDE* in epigenetic mechanisms, it appears possible that the expression of this gene and its methylation create a feedback loop of some kind, making their mutual correlation hard to catch. Another plausible explanation of the discussed phenomenon could be that microarrays used in the present study gave us the opportunity to assess just a single CpG site (GRCh37:chr16:54,962,832, being one of 9 CpGs forming the identified DMR), out of a large CpG island, GRCh37:chr16:54,962,540-54,962,906. This island, encompassing the promoter and the first exon of *CRNDE*, consists of 367 bp and contains 63 CpG sites. In our preliminary experiments on some OvCa samples, we managed to amplify and analyze this entire region by methylation-specific PCR followed by Sanger sequencing and proved the existence of methylated cytosines within this region, that are not covered by microarrays used herein (Appendix A). Thus, the *CRNDE* promoter’s methylation pattern investigated in the present study is limited to some, maybe not fully representative CpG sites, due to the relatively low density of the applied microarrays in this locus. To unambiguously determine whether there is a relationship between *CRNDE* expression and its promoter/first exon methylation, not-yet-invented techniques of a whole methylome analysis, offering much higher resolution, have to be applied. Alternatively, targeted sequencing of the *CRNDE* gene and its promoter, after bisulfite DNA conversion, could be carried out.

The results presented herein are consistent with one another, portraying the CRNDEP micropeptide as a valuable potential ovarian cancer biomarker, the clinical usability of which is probably extendable to other malignancies, as well. Moreover, this study adds to the knowledge on the *CRNDE* gene, demonstrating the positive impact of *CRNDE* promoter/first exon up-methylation on hgOvCa response to chemotherapy, concomitantly revealing the connection between such a response and the TP53 protein accumulation. Finally, we unveiled a potential prognostic value of the rs115515594 SNP in *CRNDE* in BOTS, though a molecular mechanism by which the given genetic variant affects prognosis still needs to be discovered. Thus, our study not only answers some vital *CRNDE*-related questions but also simultaneously paves the way for follow-up research.

## 4. Materials and Methods

### 4.1. Patients

In the present study, an ethnically uniform cohort of 225 patients of central European origin was investigated. All the patients were hospitalized at the Maria Sklodowska-Curie National Research Institute of Oncology, Warsaw, Poland in the years 1995–2015. Medical records of these patients were critically reviewed by at least two physicians. Their set of tumors included 76 BOTS (23 with and 53 without the *BRAF* V600E mutation), and 149 OvCa (10 lgOvCa, and 139 hgOvCa). The specimens were selected to meet the following criteria: adequate staging procedure according to the recommendations by the International Federation of Gynecologists and Obstetricians (FIGO), tumor tissue from the first laparotomy available, availability of clinical data including patient age and follow-up, as well as tumor histological type and grade, clinical stage, and residual tumor size. All tumors were uniformly histopathologically reviewed and re-classified according to new WHO criteria [42,43]. Additionally, a complete evaluation of genetic variants in the *TP53* gene (for all tumors) and the TP53 protein status (for cancers only) was performed by either next-generation sequencing (NGS, see Section 4.4) or with the mouse monoclonal antibody (Sigma-Genosys, Cambridge, UK, Appendix A), as described previously [23]. The majority of BOTS patients (N = 60) did not undergo any chemical treatment. The remaining individuals suffering from BOTS (N = 16) received chemotherapy, administered either pre- or postoperatively. All malignant tumors were excised from previously untreated patients. Thirty-five of the ovarian cancer patients were treated postoperatively with PC, while 112 underwent TP treatment after a surgical intervention. As to the evaluation of clinical endpoints, all surviving patients had at least a 3-year follow-up. In BOTS, RFS and the presence of microinvasions or implants within the tumor masses were used as dependent variables determining the disease prognosis. Whether any chemotherapy was administered to a BOTS patient or not was used as an independent binary factor in the multivariable statistical analyses. Other independent variables taken into account in the multivariable statistical inference in BOTS were the histological type of the tumor, its clinical stage according to FIGO, binary variables determining if it was primary or not, and whether it harbored the *BRAF* V600E mutation (only when the clinical impact of the rs115515594 SNP was evaluated), and also the patient age (continuous variable). In addition, BOTS were analyzed in the entire cohort of patients, and subgroups comprising specimens with (BOT V600E) or without (BOT) the *BRAF* V600E mutation, since the presence of this genetic alteration was identified here as significantly correlated with the lower age of patients diagnosed with BOTS (Wilcoxon rank sum test *p* = 0.00036, Appendix A). In cancers, OS and DFS were used as dependent prognostic variables, while PS and CR served as dependent factor variables predictive of response to treatment. CR was defined as the disappearance of all clinical and biochemical symptoms of ovarian cancer assessed after completion of the first-line chemotherapy and confirmed four weeks later. DFS was assessed only for the patients who achieved a CR. As to the independent variables used in multivariable statistical analyses in cancers, a histological type and a FIGO stage of the tumors along with the residual tumor size were taken into account as factor covariates. Of note, due to the small size of the lgOvCa subgroup, only hgOvCa samples were subjected to regression analyses performed in the present study. In such analyses, OvCa were investigated either as the entire group of specimens or in subgroups depending on the chemotherapy regimen used (PC or TP) and/or the TP53 accumulation status.

The entire set of 225 tumors had sequence variants within the coding regions of the *CRNDE* and *TP53* genes analyzed. In addition, a subset (169 snap-frozen samples only) of 21 BOTS (including 5 and 16 samples with and without the *BRAF* V600E mutation, respectively, Appendix A) and 148 OvCa (including 9 lgOvCa and 139 hgOvCa, Appendix A) had the expression of CRNDEP assessed. In another subset (153 serous tumors only) of 48 BOTS (including 21 and 27 samples with and without the *BRAF* V600E mutation, respectively), and 105 OvCa (including 9 lgOvCa and 96 hgOvCa), the DNA methylation analysis within the *CRNDE* gene was performed. Of note, clinicopathological data were missing for two BOTS and two OvCa samples. Therefore, the relevant cohorts described in Appendix A are smaller.

### 4.2. Investigating the Expression of CRNDEP

First, all tumors analyzed herein had the scc assessed by hematoxylin and eosin staining twice, right before and immediately after cryostat sections for the preparation of protein lysates were collected. The scc had to be lower than 30% for the sample to be qualified for the CRNDEP expression analysis. Total protein lysates were obtained by incubating comparable amounts of tumor cryostat sections with 300–500 µL of the RIPA buffer supplemented with the Halt Protease Inhibitor Cocktail (Thermo). The lysis was carried out until each tumor sample was dissolved. Next, the concentration of each lysate was evaluated with the BCA assay (Thermo), using bovine serum albumin (BSA, Thermo) in amounts ranging from 0 to 25 µg per well as a standard curve. Following the measurements of total protein concentration in every protein lysate, 10 µg of each lysate was transferred to the 0.22 µm pore size UltraCruz Nitrocellulose Pure Transfer Membrane (Santa Cruz Biotechnology Inc., Dallas, TX, USA) by using the Bio-Dot 96-well Microfiltration Apparatus (Bio-Rad, Hercules, CA, USA). Subsequently, to assess the levels of CRNDEP in the protein lysates, dot blot membranes were incubated with our custom-made primary rabbit anti-CRNDEP antibody for 2–3 h at room temperature. Notably, the specificity of this antibody has been confirmed in this and our previous research by both immunohistochemical staining (with the use of a blocking peptide) and Western blot, revealing a single band per lane for ectopically overexpressed CRNDEP fused to different tags (Figure 5B,C herein, and Figures 4A,B and 2E in [4]). The primary antibody was then identified with the secondary goat anti-rabbit antibody, coupled with HRP (Thermo, Appendix A), followed by detecting the enzymatic activity with the GE Healthcare Amersham™ ECL Prime Western Blotting Detection Reagent (Thermo) on the UVP ChemStudio system (Analytik Jena, Jena, Germany). The subsequent densitometric analysis was performed in the ImageJ app (v. 1.52r), supplemented with the Protein Array Analyzer plugin (v. 1.1.c). To ensure the comparability of measurements between the dot blot membranes, not only the BSA standard curve was used, but also protein lysates from the same two OvCa samples, characterized by either moderate or strong CRNDEP expression, were added to each membrane to serve as calibrators (Appendix A).

### 4.3. Analysis of Genetic and Epigenetic Alterations within the CRNDE Gene

To investigate genetic alterations in the *CRNDE* gene, the SeqCap EZ Enrichment System (Roche) was utilized to enrich genomic DNA, extracted from our entire set of ovarian tumors, in exonic regions of *CRNDE*, *TP53*, and 42 other genes associated with the development of OvCa (these results will be presented in detail in a separate publication by Szafron LA. et al.). To identify genetic variants in *CRNDE*, a previously described multi-tool bioinformatics pipeline [44], utilizing the Ensembl Variant Effect Predictor app (v. 100), was applied. Next, to perform further bioinformatic analyses, statistical inference, and outcome visualization, two R scripts, vep.r (v. 2.2) and vep.comparison.r (v. 2.2), were used. Both these programs, developed by LMS, are downloadable from https://github.com/lukszafron (LMS_gh). Finally, to interrogate linkage disequilibrium in the European population (1000 Genomes Project phase3 release V3+, N = 1006 [45]) between a single SNP in *CRNDE*, identified as a potential biomarker in BOTS, and other likely functional variants located nearby, the LDproxy application from the LDlink suite (v. 5.6.5) [46] was used.

The methylation of DNA within the entire *CRNDE* gene, including its promoter region was assessed with the use of Infinium MethylationEPIC v1.0 BeadChip microarrays (Illumina, San Diego, CA, USA). Out of 21 CpGs located within the *CRNDE* gene on the same DNA strand (minus) as the coding sequence of the gene, detectable with these microarrays, the following 16 passed all the quality filters and were analyzed in the present study (cg06119400, cg18262569, cg13471560, cg04941630, cg12599700, cg18541841, cg13499711, cg18424968, cg08440920, cg03482123, cg01686920, cg18683774, cg12719461, cg09641689, cg01901262, cg09492451). Before hybridization to a microarray, the quality of DNA was assessed with an in-house-developed method based on the comparison of real-time quantitative PCR efficiency for two amplicons of different lengths [47]. Next, 200–1000 ng of genomic DNA (the concentration measurements were performed on Qubit 4 Fluorometer with the use of Qubit™ dsDNA HS Assay Kit, both manufactured by Thermo) were bisulfite-converted with the use of the EZ DNA Methylation Kit (Zymo Research, Irvine, CA, USA). Hybridization of converted DNA to the microarrays was carried out according to the protocol provided by Illumina. The fluorescence signal was then analyzed with the iScan array scanner (Illumina). Subsequent bioinformatic analyses were performed in the R environment in line with the workflow published by Maksimovic et al. [48], which was further improved by our team to enable the analysis of methylation changes within different subregions of each gene (using functional annotations of the human genome provided by the FANTOM database (https://fantom.gsc.riken.jp/, accessed on 20 November 2021)). The differential methylation analysis of regions offered by the DMRcate package for R was also supplemented with the analysis of only those CpG sites that are located on the same DNA strand as the gene of interest. The visualization of the data was improved, as well, by introducing the hybridization quality plots presenting different percentiles of detection *p*-values, and heatmaps showing distances in methylation patterns between the analyzed samples. The R script allowing for the automation of the entire workflow (methyl_arrays.r, v. 1.2) is available for download at LMS_gh. It is worth noting that, in contrast to our NGS analysis, methylation data described herein were not mapped to the GRCh38 reference assembly of the human genome, but to its earlier version, GRCh37. Thus, the numeration of genomic locations of genetic variants reported herein always corresponds to the GRCh38 (hg38) assembly, whereas methylation changes are compliant with the GRCh37 (hg19) version of the human genome assembly.

To validate our microarray data, a single CpG site, differentially methylated between the analyzed groups of samples, was investigated by methylation-specific PCR (using the AmpliTaq Gold™ DNA Polymerase, Thermo) and Sanger sequencing with the BigDye Terminator v.3.1 Cycle Sequencing Kit (Thermo). The sequences of primers utilized in the PCR (CRNDEmF4 and CRNDEmR4) and sequencing (CRNDEmF4) reactions, along with the thermocycling conditions used, can be found in Appendix A. Finally, the methylation data were submitted to the Gene Expression Omnibus (GEO) database (data acc. no. GSE262765).

### 4.4. Evaluating the Expression of the CRNDEP-Coding Transcript in OvCa Patients

The levels of the *CRNDE* transcript encoding the CRNDEP micropeptide were assessed for 85 hgOvCa and 7 lgOvCa samples, all of which had their *CRNDE* methylation patterns and genetic variants, as well as the CRNDEP expression assessed in the present study. This gene expression evaluation was carried out with real-time quantitative PCR utilizing an in-house designed TaqMan assay, specific to the transcript of interest, according to the protocol described in our previous study [6].

### 4.5. Statistical Analyses

All statistical analyses were carried out in the R environment (versions: 3.6.1 and 4.2.3). Survival analyses were performed using the multivariable Cox proportional hazards models (package: survival, v. 3.5.5). All Cox models were also checked for the proportionality of hazards for each variable used. The prediction of treatment response was carried out by generating multivariable logistic regression models (packages: stats, v. 3.6.1 and rms, v. 6.4.1). To check the discriminating capabilities of the multivariable Cox and logistic regression models, we not only verified the statistical significance of each main variable used in such models by developing their univariable counterparts, but we also performed cross-validation of all the models in new data sets, generated from the original data by bootstrapping (with replacement) and subsequent comparison of areas under ROC curves (AUCs) for the original and bootstrapped data sets, using the riskRegression package for R (v. 2023.2.1). The R script written to automate the above-mentioned statistical inference and subsequent visualization of the results (regression.analyses.r, v. 1.1) is downloadable from LMS_gh.

## 5. Conclusions

This paper demonstrates the clinical significance of the CRNDEP micropeptide in ovarian tumor patients. The CRNDEP level was significantly elevated in highly aggressive tumors (hgOvCa) compared to benign neoplasms (BOTS without the *BRAF* V600E mutation). Consistently, high levels of this micropeptide emerged as a negative prognostic and predictive factor in hgOvCa. The cancer-promoting role of CRNDE(P), demonstrated in our recent paper [5], was also supported by our genetic and epigenetic studies carried out herein. We found no CRNDEP-disrupting mutations in our set of ovarian tumor samples and showed hypomethylation of the *CRNDE* gene in more aggressive tumors. Accordingly, down-methylation of the *CRNDE* promoter increased hgOvCa resistance to chemotherapy in the case of tumors with TP53 protein accumulation. Moreover, we established that the rs115515594 SNP in *CRNDE* is a potential, negative prognostic factor in BOTS only, in which this genetic variant was highly overrepresented compared to healthy individuals. Concomitantly, the same polymorphism turned out to be more frequent in BOT V600E tumors than in hgOvCa. Taken together, these results contribute to a better understanding of the regulatory mechanisms the *CRNDE* gene and its protein product are involved in. Our study may also add to the improvements in OvCa screening and diagnosis, and conceivably lead to the development of less aggravating and more effective methods for the future molecular therapy of this disease.

## Figures and Tables

**Figure 1 ijms-25-07531-f001:**
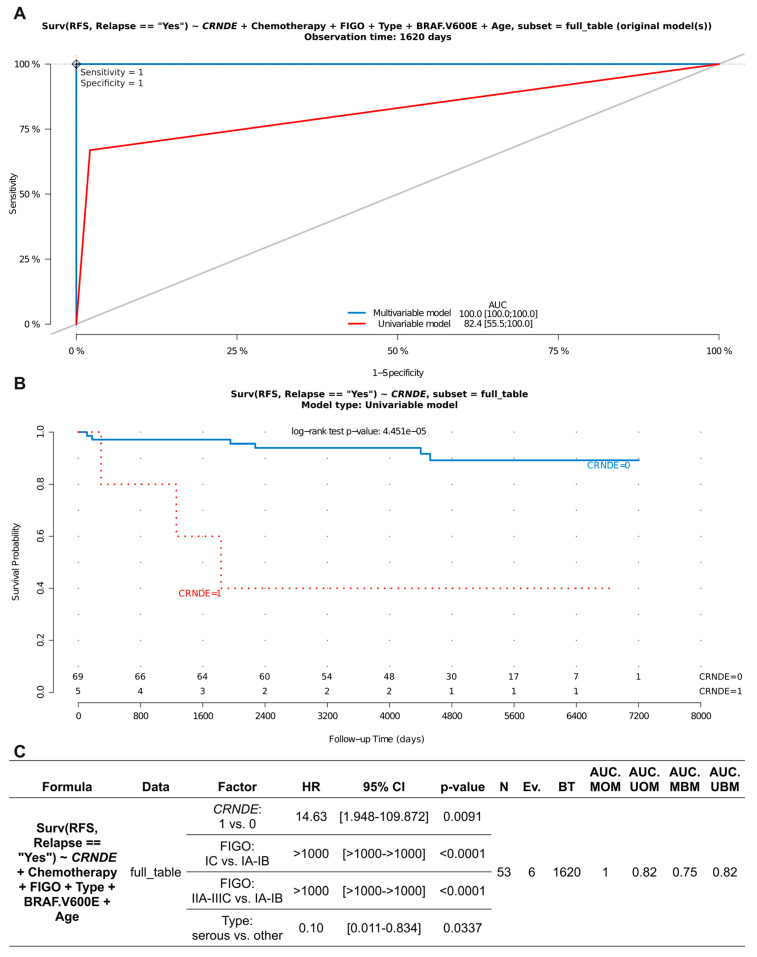
Assessment of discriminating capabilities of the rs115515594 SNP in the CRNDE gene in BOTS. The ROC curves plotted for the multivariable and univariable Cox regression models (**A**) show that this SNP can serve as a predictor of poor prognosis (increasing the risk of relapse) in BOTS patients. The corresponding Kaplan–Meier survival curves (**B**), plotted for the univariable Cox model (presence (1) vs. absence (0) of the given CRNDE variant), demonstrate its discriminating value. The detailed outcome (significant results only) of the multivariable Cox regression analysis is shown in (**C**). RFS—relapse-free survival; HR—hazard ratio; CI—confidence interval; Ev.—events no.; AUC—area under a ROC curve; MOM—multivariable original model; UOM—univariable original model; MBM—multivariable bootstrapped model; UBM—univariable bootstrapped model; BT—best time (follow-up time in days with the highest AUC value for the MBM); Lower number of observations (N) in C than in B results from the fact that non-primary BOTS were excluded from the multivariable analysis, since FIGO stages were not assessed for such tumors. Low *p*-values are displayed in exponential notation (e−n), in which e (exponent) multiplies the preceding number by 10 to the minus nth power.

**Figure 2 ijms-25-07531-f002:**
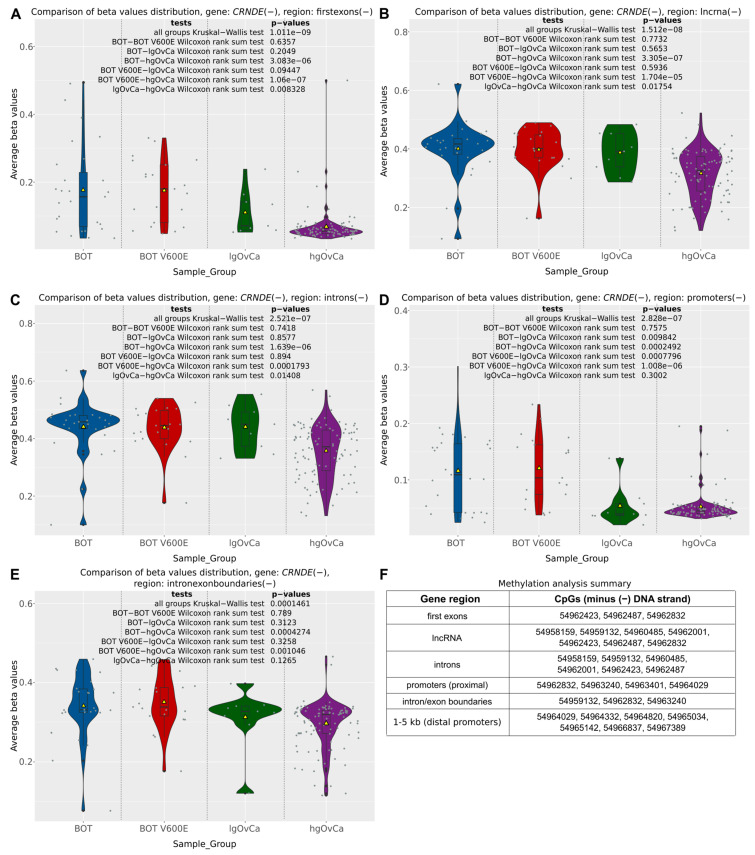
CpG methylation alterations in the *CRNDE* gene. Violin plots supplemented with boxplots showing differences in methylation patterns (beta values) in five different regions of the *CRNDE* gene (first exon (**A**), lncRNA (**B**), introns (**C**), proximal promoter (**D**), intron/exon boundaries (**E**) between BOTS without (BOT) and with the *BRAF* V600E mutation (BOT V600E), lgOvCa, and hgOvCa. Yellow triangles in the boxplots represent mean values, while gray horizontal bars denote medians. The results of the Kruskal–Wallis and Wilcoxon rank sum tests are also provided. In Figure (**F**), genomic locations (on GRCh37:chr16) of CpG sites included in each region are listed. All the CpG sites are located on the same DNA strand (minus (−)) as the *CRNDE* gene. The plots for the distal promoter (1–5 kb) were not included, as in this region, there were no statistically significant differences in methylation patterns between the analyzed groups of tumors. Low *p*-values are displayed in exponential notation (e−n), in which e (exponent) multiplies the preceding number by 10 to the minus nth power.

**Figure 3 ijms-25-07531-f003:**
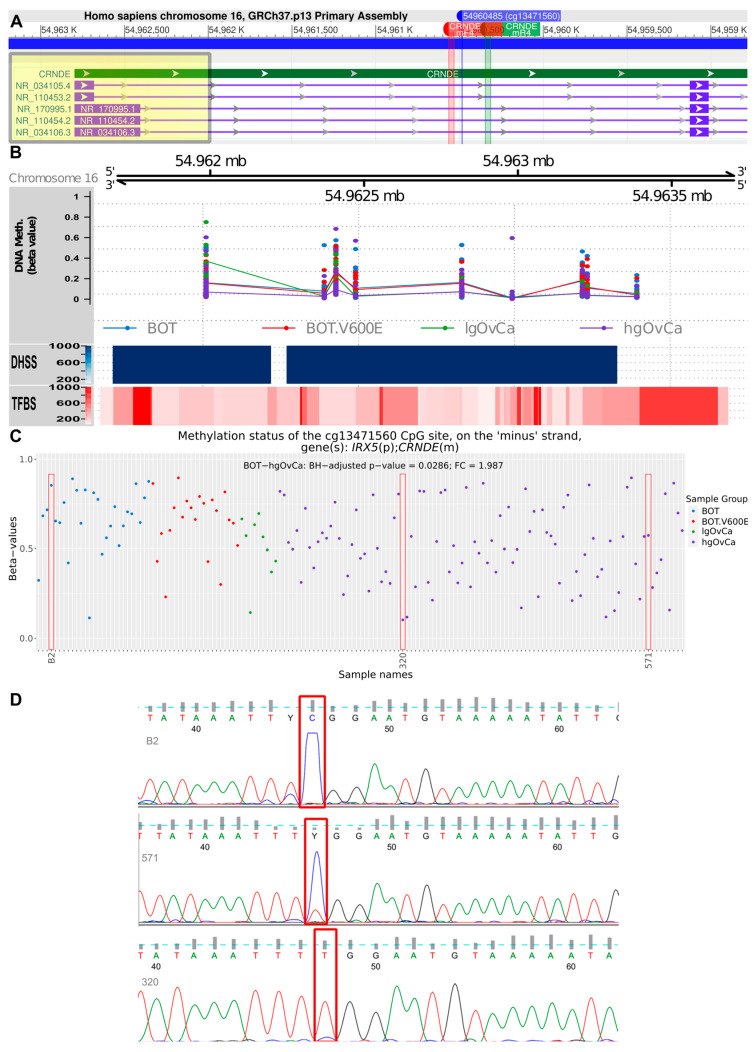
Assessment of the *CRNDE* gene methylation. (**A**): Fragment of the *CRNDE* gene comprising the differentially methylated region (DMR, GRCh37:chr16:54962001-54963401), and locations of two primers, CRNDEmF4 and CRNDEmR4, used to verify methylation patterns in one CpG site (cg13471560, GRCh37:chr16:54960485) by PCR and Sanger sequencing. The DMR is highlighted in yellow, while the PCR primers are marked red and green. A detailed visualization of the DMR region is shown in (**B**). Apart from chromosomal localization and DNA methylation levels (beta-values), the DHSS (Dnase I hypersensitive sites) and TFBS (transcription factor binding sites) are also displayed in this plot. In (**C**), differences in methylation level (beta values) for the cg13471560 CpG site are presented in a dot plot, supplemented with significant results of linear regression modeling (only the BOT without BRAF V600E mutation vs. hgOvCa comparison). Red rectangles indicate three samples, which had their methylation levels validated by methylation-specific PCR and Sanger sequencing. In the sequencing results (**D**), the cytosine forming the relevant CpG site is marked with red rectangles. BH—Benjamini–Hochberg correction for multiple comparisons. Y in (**D**) stands for pyrimidine (C or T) according to IUPAC nucleotide codes.

**Figure 4 ijms-25-07531-f004:**
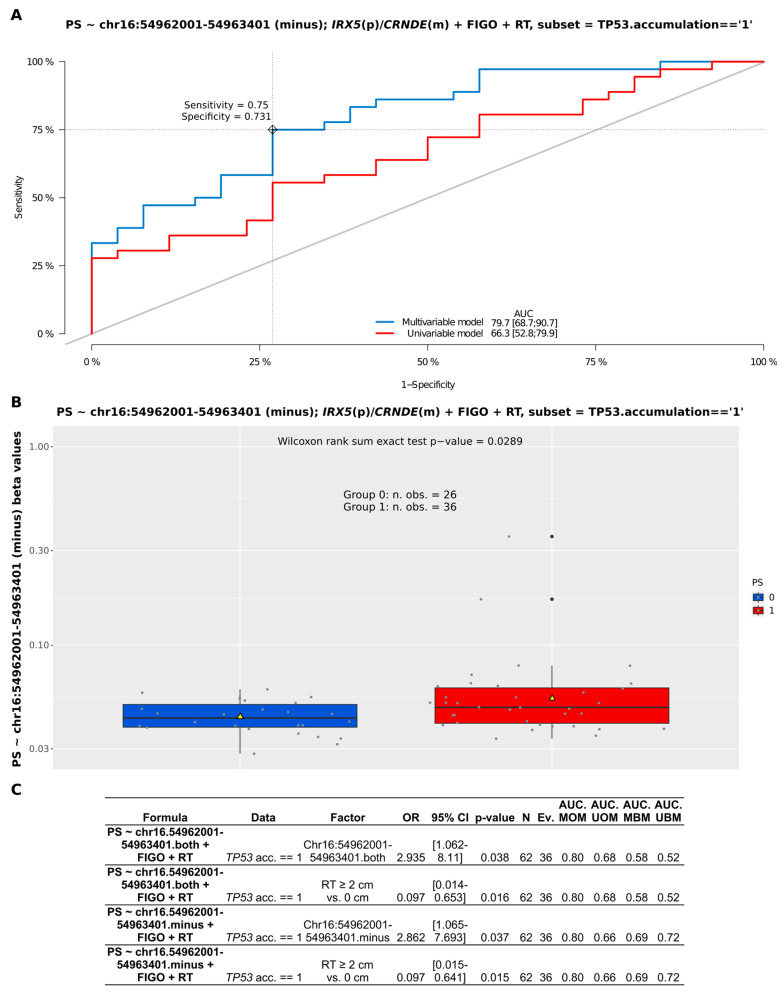
Assessment of discriminating capabilities of methylation changes (beta values) in the DMR (GRCh37:chr16:54962001-54963401) within the *CRNDE* gene in patients with serous hgOvCa exhibiting TP53 accumulation in the tumors. The ROC curves plotted for the multivariable and univariable logistic regression models (**A**) show that methylation changes in the analyzed locus can serve as a predictor of hgOvCa sensitivity to chemotherapy (PS). In the corresponding boxplot (**B**), PS is categorized as either 0 (resistant) or 1 (sensitive) to treatment. The boxplot is supplemented with mean values (yellow triangles) and the result of the Wilcoxon rank sum test. The detailed outcome (significant results only) of the multivariable logistic regression analyses for the given DMR (comprising CpGs located either on both DNA strands or on the minus strand only) is shown in (**C**). DMR—differentially methylated region; m—minus DNA strand; *p*—plus DNA strand; RT—residual tumor size; TP53.acc. == 1—presence of the TP53 protein accumulation; OR—odds ratio; CI—confidence interval; Ev.—events no.; AUC—area under an ROC curve; MOM—multivariable original model; UOM—univariable original model; MBM—multivariable bootstrapped model; UBM—univariable bootstrapped model.

**Figure 5 ijms-25-07531-f005:**
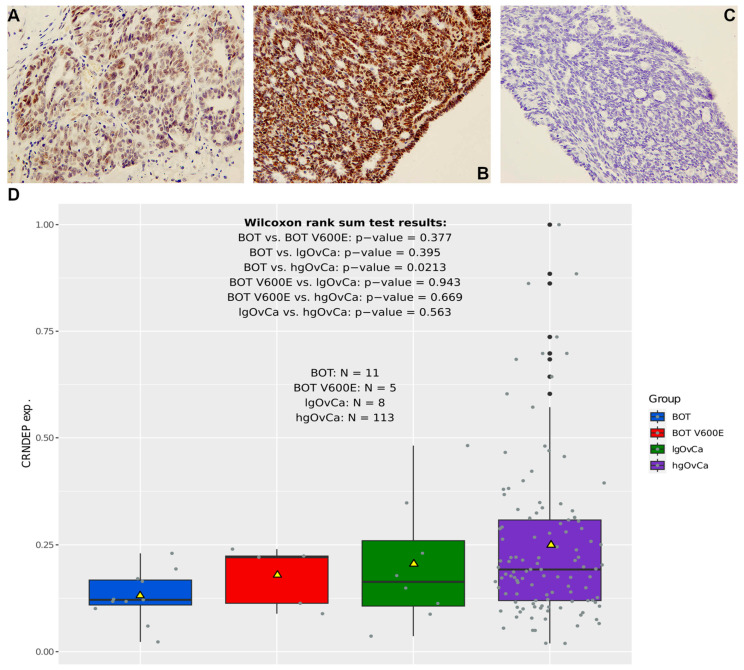
CRNDEP expression analysis in ovarian tumors. (**A**–**C**): exemplary IHC results depicting moderate (**A**) and strong (**B**) expression of CRNDEP in two ovarian tumors. Negative IHC staining in the “B” tumor after utilization of a blocking peptide is shown in (**C**). Pictures in (**A**–**C**) were all taken under 200× magnification. In (**D**), boxplots presenting differences in CRNDEP expression in serous ovarian tumors of different aggressiveness are shown. The plots are supplemented with the results of the Wilcoxon rank sum tests. Yellow triangles represent mean values.

**Figure 6 ijms-25-07531-f006:**
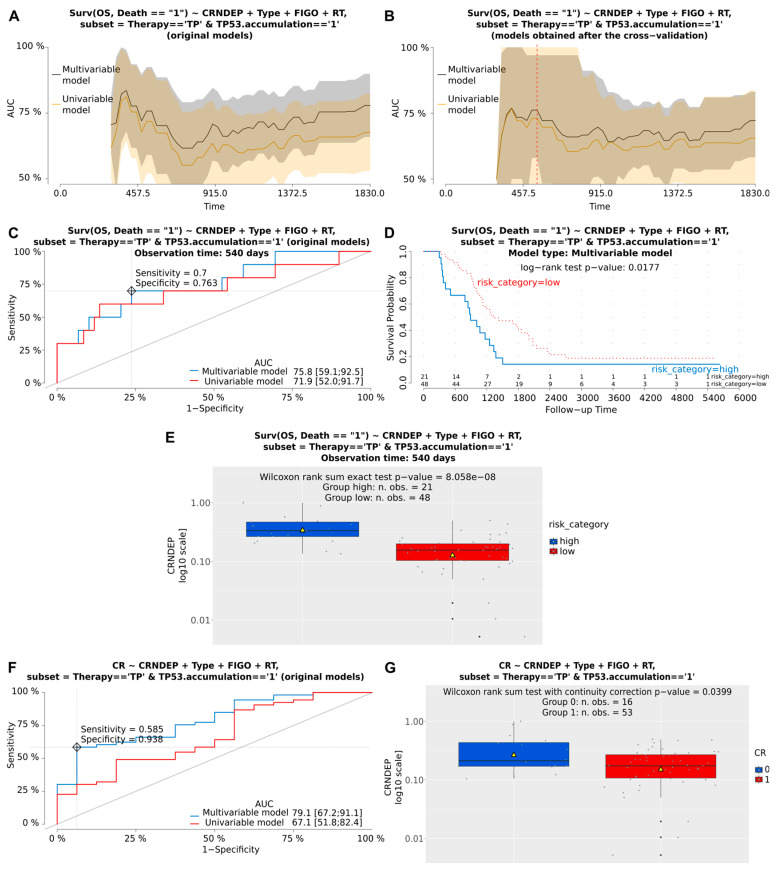
Visualization of exemplary results of the Cox and logistic regression analyses. These results depict the prognostic and predictive meaning of altered CRNDEP expression in hgOvCa patients, treated with the taxane/platinum (TP) therapy, whose tumors exhibited the TP53 protein accumulation. Plots demonstrating the prognostic value of CRNDEP (estimation of overall survival (OS) and the risk of death) or its ability to predict complete remission (CR) are shown in (**A**–**G**), respectively. AUC plots for original uni- and multivariable models (**A**) and the corresponding models after bootstrap-based cross-validation of the set (**B**). A red dashed line in B indicates the same time point that was used to draw the time-dependent ROC curves (**C**). Optimal cutoff points in ROC curves (**C**,**F**) were calculated for the multivariable models using the Youden method. Discrimination sensitivity and specificity values for the given cutoff points are also provided in (**C**,**F**,**D**): Kaplan–Meier survival curves obtained for the patients divided into two categories (risk higher (high) or lower (low) than for the ROC curve (**C**)-estimated cutoff point, based on the risk of death calculated using the multivariable model). The Kaplan–Meier curves are supplemented with the result of the log-rank test as well. (**E**,**G**): boxplots showing CRNDEP expression changes in patients with a high or low risk of death (**E**), and with (1) or without (0) CR (**G**). Yellow triangles in boxplots represent mean values. Type—histological type; RT—residual tumor size. Low *p*-values are displayed in exponential notation (e−n), in which e (exponent) multiplies the preceding number by 10 to the minus nth power.

**Table 1 ijms-25-07531-t001:** Multivariable Cox and logistic regression analyses of CRNDEP expression in hgOvCa patients.

Formula	Data	Factor	HR/OR	95% CI	*p*-Value	N	Ev.	BT	AUC.MOM	AUC.UOM	AUC.MBM	AUC.UBM
**Surv(OS, Death == 1) ~ CRNDEP + Type + FIGO + RT**	All	CRNDEP	3.217	[1.1–9.2]	0.0291	138	120	1530	0.80	0.65	0.75	0.64
		FIGO: IIIA-B vs. ≤IIC	9.516	[1.2–75]	0.0319	138	120	1530	0.80	0.65	0.75	0.64
		FIGO: IV vs. ≤IIC	10.072	[1.3–80]	0.0291	138	120	1530	0.80	0.65	0.75	0.64
		RT: <2 cm vs. 0 cm	2.382	[1.4–4.1]	0.0014	138	120	1530	0.80	0.65	0.75	0.64
		RT: ≥2 cm vs. 0 cm	3.008	[1.7–5.5]	0.0003	138	120	1530	0.80	0.65	0.75	0.64
	TP53.acc. == 1	CRNDEP	5.458	[1.0–29]	0.0450	92	77	420	0.76	0.74	0.73	0.76
		RT: <2 cm vs. 0 cm	2.483	[1.3–4.8]	0.0060	92	77	420	0.76	0.74	0.73	0.76
		RT: ≥2 cm vs. 0 cm	3.457	[1.7–7.0]	0.0006	92	77	420	0.76	0.74	0.73	0.76
	TP	CRNDEP	6.388	[2.0–21]	0.0020	107	90	450	0.77	0.68	0.73	0.66
		FIGO: IIIA-B vs. ≤IIC	9.861	[1.2–83]	0.0351	107	90	450	0.77	0.68	0.73	0.66
		FIGO: IIIC vs. ≤IIC	8.258	[1.1–63]	0.0414	107	90	450	0.77	0.68	0.73	0.66
		FIGO: IV vs. ≤IIC	10.598	[1.3–90]	0.0304	107	90	450	0.77	0.68	0.73	0.66
		RT: <2 cm vs. 0 cm	2.270	[1.2–4.1]	0.0075	107	90	450	0.77	0.68	0.73	0.66
		RT: ≥2 cm vs. 0 cm	3.162	[1.6–6.3]	0.0011	107	90	450	0.77	0.68	0.73	0.66
	TP & TP53.acc. == 1	CRNDEP	9.409	[1.6–55]	0.0128	69	55	540	0.76	0.72	0.76	0.73
		RT: <2 cm vs. 0 cm	2.729	[1.3–6.0]	0.0117	69	55	540	0.76	0.72	0.76	0.73
		RT: ≥2 cm vs. 0 cm	3.182	[1.4–7.4]	0.0072	69	55	540	0.76	0.72	0.76	0.73
**CR ~ CRNDEP + Type + FIGO + RT**	All	CRNDEP	0.564	[0.4–0.9]	0.0067	138	96	NA	0.74	0.65	0.69	0.65
		RT: ≥2 cm vs. 0 cm	0.152	[0.0–0.7]	0.0110	138	96	NA	0.74	0.65	0.69	0.65
	TP53.acc. == 1	CRNDEP	0.482	[0.2–0.9]	0.0332	92	67	NA	0.73	0.65	0.60	0.66
		RT: ≥2 cm vs. 0 cm	0.169	[0.0–1.0]	0.0478	92	67	NA	0.73	0.65	0.60	0.66
	TP	CRNDEP	0.531	[0.3–0.8]	0.0053	107	78	NA	0.77	0.66	0.68	0.64
		RT: ≥2 cm vs. 0 cm	0.168	[0.0–1.0]	0.0446	107	78	NA	0.77	0.66	0.68	0.64
	TP & TP53.acc. == 1	CRNDEP	0.368	[0.2–0.9]	0.0230	69	53	NA	0.79	0.67	0.70	0.65
**PS ~ CRNDEP + Type + FIGO + RT**	All	CRNDEP	0.603	[0.4–0.9]	0.0162	138	82	NA	0.74	0.63	0.68	0.61
		RT: ≥2 cm vs. 0 cm	0.132	[0.0–0.5]	0.0017	138	82	NA	0.74	0.63	0.68	0.61
	TP	CRNDEP	0.591	[0.4–0.9]	0.0160	107	69	NA	0.74	0.62	0.68	0.63
		RT: ≥2 cm vs. 0 cm	0.130	[0.0–0.6]	0.0090	107	69	NA	0.74	0.62	0.68	0.63

OS—overall survival, Type—histological type; RT—residual tumor size; CR—complete remission; PS—platinum sensitivity; TP53.acc. == 1—presence of the TP53 protein accumulation; TP—taxane/platinum regimen; HR—hazard ratio, OR—odds ratio; CI—confidence interval; Ev.—events no.; AUC—area under an ROC curve; MOM—multivariable original model; UOM—univariable original model; MBM—multivariable bootstrapped model; UBM—univariable bootstrapped model; BT—follow-up time in days with the highest AUC value for the MBM.

## Data Availability

All data are available in the main text or the Appendix A.

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
