# Peer review of "The Clinical Significance of CRNDE Gene Methylation, Polymorphisms, and CRNDEP Micropeptide Expression in Ovarian Tumors"

_ijms, 2024, doi:10.3390/ijms25147531_

Round 1

Reviewer 1 Report

Comments and Suggestions for Authors

The manuscript titled “Clinical significance of the CRNDE gene methylation, polymorphisms and the CRNDEP micropeptide expression in ovarian tumor patients“ by Laura Aleksandra Szafron et al.

Here are my concerns:

- The introduction part is too brief, it should be elaborated and the authors introduce more on the strucure of CRNDE lncRNA, the protein(s) encoded. So a figure should be designed. Also, the amino acid sequences of this micropeptide should be shown.

- A larger cohort size should be included.

- The ethnicity of the patients should be shown.

- Would this micropeptide be detected in blood?

- It would be necessary to include a summary figure to conclude and present the essence of the findings in this work.

Comments on the Quality of English Language

Typos and unfriendly mode of English usage can be found.

Reviewer 2 Report

Comments and Suggestions for Authors

This is a pioneering study assessing  CRNDEP micropeptide polymorphism and methylation in association with Ovarian tumor patients. 

However, current understanding of genetic predisposition to OvC is polygenic risk score format. We have numerous other factors that, cumulatively. To ascertain the actual impact of CRNDEP alone or in augmenting another factor, there is a need to explore other factors to confidently derive the conclusions. There are numerous gene polymorphisms reported e.g. ATM, BRCA1, BRCA2, CHEK2, PALB2 (FANCN), PTEN, TP53, and CDH1 for OvC. As a negative control known genes with insufficient association can be included e.g. BRD1, BRIP1, RAD51C, and RAD51D.

The reported association of one of the SNP with poor prognosis is moderately significant, and to defend claims to use this gene as a biomarker requires an objective comparison and association with other known factors. 

OvC has 50% risk from BRCA1, 15% from BRCA2, 10% from BRIP1 and 15% from RAD51C... 

CRNDEP alone is comparable to ATM and PALB2.. With so many factors and unknown LD with CRNDEP it's wrong to assume the predisposition potential

Comments on the Quality of English Language

Minor language corrections needed. The claims are phrased too boldly, a more scientific tone is needed in abstract and result sections. 

Reviewer 3 Report

Comments and Suggestions for Authors

In manuscript ijms-3010864 the authors describe possible roles of the lncRNA CRNDE and its encoded micro-peptide (CRNDE(P)) in patients affected by ovarian tumor of different degrees of severity. They highlight a different methylation pattern of the lncRNA-encoding gene in different tumors and its relationship with TP53 mutations, and a positive relationship between micro-peptide expression and tumor grade. This work is an extension of previous works, either by the same group or from other labs, describing this locus and its importance in carcinogenesis. Overall, the manuscript is well written, the experiments are convincing, and the scientific background is robust. Though, some points should be fixed before any decision on publication.

1.      The authors affirm that CRNDE(P) has a direct role in carcinogenesis. What I see in this manuscript is a positive correlation, but not a cause-effect setting. Thus, I believe that the authors should better explain how they see this direct involvement; I do not see experiments aiming at depleting/boosting the expression of CRNDE or CRNDE(P) to see cell reactions to external stimuli (e.g., chemotherapy). Alternatively, the following sentences should be changed to underline this fact. Lines 173-174: “high CRNDEP expression diminished the chances of both hgOvCa tumors’ complete remission (CR), and their sensitivity to chemotherapy (PS)” (I don’t see data showing that CRNDE(P) directly acts on remission or chemo-resistance, I just see that resistant cells have higher micro-peptide levels); lines 304-305: “proved that its protein product, CRNDEP, elicits a similar, cancer-driving effect” (I don’t see data showing how CRNDE(P) “drives” cancer development, I just see a positive correlation, useful as a biomarker, but nothing more); lines 532-535: “The cancer-promoting role of CRNDE(P) was also supported by our genetic and epigenetic studies, revealing no CRNDEP-disrupting mutations in our set of ovarian tumor samples and showing hypomethylation of the CRNDE gene in more aggressive tumors” (also here, there is no evident data showing that; moreover, the increasing level of the peptide in more aggressive tumors might be a byproduct of altered gene methylation pattern, and this does not prove per se a cause-effect relationship).

2.      Lines 288-290: the authors write that “increased methylation of the CRNDE DMR (probably leading to decreased CRNDE expression) was associated with higher sensitivity of cancer cells to chemical treatment”. The authors do not prove that, indeed, this methylation affects CRNDE expression, which is strange, since I think they should have the possibility and expertise to make a measurement (even an approximate estimation would suffice, if data are evident) of gene expression in tumors and controls. Without a quantitative analysis, the authors should only affirm that there is a relationship between methylation pattern and aggressiveness (another useful biomarker, but nothing more). I invite the authors either to perform the gene expression quantification, or to highlight in the discussion and conclusions that (again) no cause-effect condition can be assessed in this experimental setup.

3.      Some figures are difficult to read, at least for me. For example, yellow triangles in figure 2 (panels a-b-c-d-e), figure 4b and figure 6 (panels e-g) are barely visible; gray dots over green plots (both violin and box plots in all figures) and those in figure 4b and 6e/6g (blue plots) are barely visible; in some cases, dots on violin plots cannot be easily assigned to the left or right plot (see for example figure 2a, BOT vs. BOT V600E).

Round 2

Reviewer 2 Report

Comments and Suggestions for Authors

Authors have meticulously addressed all the comments raised. The revised manuscript meets the publication rigor criteria.

Reviewer 3 Report

Comments and Suggestions for Authors

I think that now the manuscript is worth of publication. The authors convincingly replied to my observations and performed/discussed the requested experiments. Only a minor note on lines 434-435 (new version); in the sentence "According to the literature, the levels of gene transcripts and the corresponding genes are not always concordant", I think that the word "genes" should be replaced with "proteins".